# Pan-cancer analysis of PSCA that is associated with immune infiltration and affects patient prognosis

Chenxing Wang[1,2☯], Xingjia Zhu[1,2☯], Ming Zheng[2,3☯], Yixun Chen[4,5], Rui Jiang[1,2], Xinyi He[5], Zongheng Liu[1,5], Zhichao Lu[1,5], Ziheng Wang[5]*, Yang Yang[6]*

1 Department of Neurosurgery, Affiliated Hospital of Nantong University, Medical school of Nantong University, Nantong, China, 2 Research Center of Clinical Medicine, Affiliated Hospital of Nantong University, Medical School of Nantong University, Nantong, China, 3 Department of Laboratory Medicine, Affiliated Hospital of Nantong University, Medical School of Nantong University, Nantong, China, 4 Eye Institute, Affiliated Hospital of Nantong University, Medical School of Nantong University, Nantong, Jiangsu, China, 5 Department of Clinical Biobank & Institute of Oncology, Affiliated Hospital of Nantong University, Medical School of Nantong University, Nantong, China, 6 Department of Trauma Center, Affiliated Hospital of Nantong University, Medical School of Nantong University, Nantong, China

☯ These authors contributed equally to this work.
* wang.ziheng@connect.um.edu.mo (ZW); yangyang286228@ntu.edu.cn (YY)

**Data Availability Statement:** All relevant data are within the manuscript and its Supporting information files.

## Abstract

Prostate stem cell antigen (PSCA) is associated with disease progression, promotion of angiogenesis, invasion, metastasis and immune evasion in cancer. However, its expression pattern and diagnostic and prognostic potential have not been thoroughly analysed from a pan-cancer perspective. This study aimed to examine the effects of PSCA on the prognosis and inflammatory cell infiltration patterns of various cancer types. We analysed the relationship between PSCA expression and immunological subtypes in tumor microenvironment (TME) and the role of molecular subtypes, potentially promising immune biomarkers and tumour-infiltrating lymphocytes (TILs) in various cancer types, especially lung adenocarcinoma (LUAD). In addition, we investigated the prognostic significance of PSCA expression in LUAD. The co-expression network of PSCA was found to be mainly involved in the regulation of immune responses and antigen processing and expression and was significantly enriched in pathological and substance metabolism-related pathways in cancer. Altogether, this study reveals that PSCA is a promising target for immunotherapy in patients with cancer.

## Introduction

The PSCA gene is associated with disease progression, promotion of angiogenesis, invasion, metastasis and avoidance of immune surveillance in cancer [1,2]. For example, it has been shown that PSCA expression is elevated in pancreatic and gastric cancers. However, its expression pattern and diagnostic and prognostic potential across cancers remain elusive [3,4]. This study aimed to reveal the role of PSCA in the diagnosis and prognosis of various cancer types,

**Funding:** This study was supported by Science and Technology Project of Nantong City (MS22021044). The author YY was funded by the project. The funder's website is http://kjj.nantong.gov.cn/ The funders had no role in study design, data collection and analysis, decision to publish, or preparation of the manuscript.

**Competing interests:** The authors have declared that no competing interests exist.

especially LUAD, primarily based on The Cancer Genome Atlas (TCGA) data. In addition, dysregulation of PSCA was examined from a pan-cancer perspective. Previous studies have validated that patients with cancer tend to produce autoantibodies. Tissue-specific antigens may serve as a target for adoptive T-cell transfer-based immunotherapy. The body produces antibodies by inducing inflammation or increasing the production of self-antigens. Owing to the presence of antibodies in the early stages of the disease and the relative availability of serum samples, antibody diversity plays an important role in the diagnosis and prognosis of tumours in clinical settings. In addition, the presence of antibodies indicates the relative immunogenicity of an antigen. Therefore, analysing the response of tissue-specific antigens and their antibodies is necessary for detecting the efficacy of immunotherapy.

PSCA is expressed on the surface of basal epithelial cells [5]. As regulators of the activity of nicotinic acetylcholine receptors (nAChRs) [6]. In vitro studies have demonstrated that PSCA is downregulated in prostate cancer and gastric cancer cells [7,8]. In addition, PSCA is differentially expressed among different tumours9 and may inhibit cell proliferation. Inhibition of nicotine-induced signalling may implicate α-3:β-2– or α-7–containing nAChRs in vitro [9,10].

However, the expression pattern as well as the diagnostic and prognostic potential of PSCA have not been analysed comprehensively from a pan-cancer perspective. In this study, we examined the effects of PSCA on the prognosis and immune landscape of cancer. In addition, we investigated the potential relationship between PSCA expression and immunological subtypes of TME [11,12] and the role of molecular subtypes, promising biomarkers of immunity [13] and tumour-infiltrating lymphocytes (TILs) in various cancer types, especially LUAD [14]. In particular, we examined the effects of PSCA expression on the prognosis of LUAD. Altogether, this study highlights the role of PSCA in immunotherapy and may guide the development of novel therapeutic strategies for cancer [15,16].

## Materials and methods

### Data sources and collection

RNA expression and clinical data were extracted from The Cancer Genome Atlas (TCGA) and Genotype-Tissue Expression (GTEx) projects from the UCSC Xena database (https://xenabrowser). Data on DNA copy number and methylation were obtained from the cBioPortal database (https://www.cbioportal.org/).

### Mutation profiles

The cBioPortal for Cancer Genomics (http://www.cbioportal.org) is a repository of large-scale cancer genomics datasets. The Cancer Somatic Mutation Inventory (https://cancer.sanger.ac.uk/cosmic/) [17] is used for determining the impact of somatic mutations on cancer [18].

### Correlation analysis

PSCA expression was analysed at the chromosomal level, and its correlation with other variables was examined in various cancer types [18]. Pearson analysis was used to assess the correlation of PSCA with immune checkpoints [19] and mismatch repair (MMR) proteins [20]. The 'pheatmap' package in R was used to generate a heat map for visualising the results [15,21].

### Immune infiltration

The tumour purity of 33 human cancers was evaluated using the 'ESTIMATE' package [1,11]. The relationship between PSCA expression and tumour purity scores in various cancer types was visualised on a scatter plot. The Tumor Immune Estimation Resource 2.0 (TIMER2.0;

http://timer.cistrome.org/) web server is used for systematic analysis of immune infiltration. The differential expression of PSCA between tumour and adjacent normal tissues was analysed, and the relationship between PSCA expression and immune infiltration was examined using several immune-related deconvolution algorithms [11,22]. TISIDB [23] (http://cis.hku.hk/TISIDB/) was used to examine whether PSCA expression was different between patients who responded and did not respond to immunotherapy. In addition, we assessed the correlation between PSCA expression and markers of immune cell subsets.

## Statistical analysis

All data are expressed as the mean ± standard deviation (SD). Differences between groups were analysed using Student's t-test. In addition, the correlation in the scatter plot was evaluated using Spearman's rank correlation coefficient. Furthermore, the survival difference between groups in the survival curves were analysed by the Log-rank test. Statistical analysis was performed using the R (version 3.6.2) software. A p-value of <0.05 (two-tailed) was considered statistically significant.

## Cell culture and transient transfection

U1755, H1299, A549 and H661 cells were purchased from ATCC (Manassas, USA). All cells were cultured in Gibco DMEM/F-12 medium (Thermo Fisher Scientific, USA). The negative control (NC) siRNA and siRNAs targeting PSCA (Invitrogen, USA) were transfected into the cells using Lipofectamine 2000 (Invitrogen, USA). The sequences of siRNAs targeting PSCA were as follows: PSCA-si-1, CCGGCAGATCGGCTCTATTGACA; PSCA-si-2, GGCAGATCGGCTCTATTGACACA; PSCA-si-3, GCCCAGCATTCTCCACCCTTAAC.

## Real-time PCR

Real-time reverse transcription polymerase chain reaction (qRT-PCR) was performed as described in our previous study [24,25]. The primer sequences used for PCR are as follows:

| Gene | Forward primer (5–3) | Reverse primer (5–3) |
|------|---------------------|---------------------|
| PSCA | GAACTGCGTGGATGACTCAC | CAAGTCGGTGTCACAGCACG |
| GAPDH | AATGGGCAGCCGTTAGGAAA | GCCCAATACGACCAAATCAGAG |

## Western blotting

Tumour and normal tissues were lysed in RIPA buffer (Solarbio, China) and denatured at 100˚C for 15 min. Extracted proteins were separated on 10% sodium dodecyl sulphate—polyacrylamide gels and transferred to polyvinylidene difluoride (PVDF) membranes. The membranes were blocked with 5% skim milk for 1 h and incubated overnight with primary antibodies, including anti-PSCA (1:200, Proteintech, 17171-1-AP) and anti-GADPH (1:5000, Proteintech, 60004-1-Ig) antibodies. The following day, the membranes were incubated with secondary antibodies for 2 h at ambient temperature, and protein bands were detected using an ECL kit (Billerica Millipore, USA). This study was approved by the Human Ethics Committee of the Affiliated Hospital of Nantong University (No. 2018-K020), and written informed consent was obtained from all patients.

## Transwell assay

Transwell assay was performed to assess cell migration and invasion. Briefly, cells ($5 \times 10^4$) were inoculated in Matrigel-coated (for invasion) or uncoated (for migration) Transwell chambers. Serum-free medium (SFM) was added to the upper chamber, whereas DMEM was added to the lower chamber. After 24 hours of incubation, the cells were stained with 0.1% crystal violet and counted under a microscope.

## Flow cytometry

The cell cycle and apoptosis were analysed via flow cytometry according to the manufacturer's instructions. To examine the cell cycle, cells were harvested using trypsin and resuspended in PBS at a concentration of $1 \times 10^5$ cells/100 μL. Subsequently, the cells were stained with propidium iodide (PI) on ice for 30 min, washed with PBS and detected on a BD FACSCalibur Flow Cytometer (USA). The distribution of cells within the cell cycle was analysed using the ModFit software. For the detection of apoptosis, cells were washed, resuspended in pre-chilled PBS and stained with Annexin V-647 and PI solution in the dark. After 15 minutes of incubation, apoptosis was detected via flow cytometry.

# Results

## Expression and mutation patterns of PSCA in pan-cancer

The mutation and expression patterns of PSCA were compared among different cancers. Data on copy number variations were integrated at the level 4 gene level. The frequency of copy number variations was significantly different among GBMLGG (N = 603, loss = 18, gain = 36) (p = 0.03), CESC (N = 255, gain = 32, loss = 5) (p = 1.2e-3), BRCA (N = 856, gain = 171, loss = 56) (p = 5.6e-10), SARC (N = 212, loss = 17, gain = 28) (p = 0.05), PRAD (N = 461, loss = 19, gain = 12) (p = 0.04), PAAD (N = 158, gain = 19) (p = 9.0e-3), OV (N = 204, loss = 19, gain = 193) (p = 1.4e-6) and CHOL (N = 30, gain = 6) (p = 0.04) (Fig 1A). Based on the TCGA pan-cancer gene expression profiling analysis of PSCA expression differences in tumour tissues, the differential analysis of PSCA combined with unpaired (Fig 1B) and combined with GTEx normal paired samples (Fig 1C). The results suggested that PSCA was significantly upregulated in UCEC, BRCA and ESCA and significantly downregulated in GBM, KIRP, PRAD, STAD, HNSC, KIRC and READ. In addition, multi-omic analysis of expression data of LUAD samples validated that multiple differentially expressed genes, including PSCA, contributed to the increased risk of disease recurrence. However, the prognostic potential of PSCA in LUAD remains unclear. Therefore, we performed in-depth multi-omic analysis based on TCGA-LUAD dataset. A waterfall plot was generated to analyse the mutation pattern of PSCA in LUAD, and differences in the mutation frequency of PSCA among samples were estimated via chi-square test (Fig 1D). Subsequently, the level 4 single-nucleotide variant data of all samples in TCGA-LUAD dataset, which were processed by MuTect2, were integrated, and protein domain information was acquired using the 'maftools' R package (Fig 1E).

## Comprehensive analysis of tumour stemness index of PSCA in LUAD

We obtained six tumour stemness indices based on mRNA expression and methylation from previous studies and analysed the correlation among them20 (Fig 1F) and differences in the indices among samples (S1 Fig). The six indices were as follows: RNA-based stemness index (S1B Fig), epigenetic regulation-based stemness index (S1E Fig), DNA methylation-based stemness index (S1A Fig), epigenetically regulated DNA methylation-based stemness index

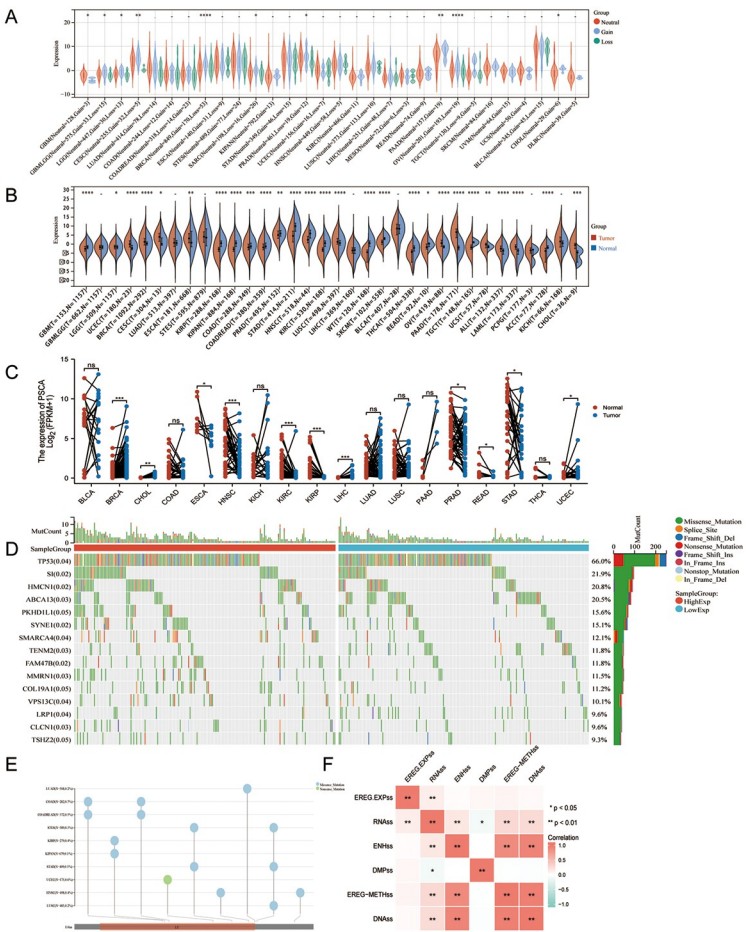

**Fig 1. Expression and mutation patterns of PSCA in pan-cancer. (A)** Violin plot demonstrating differences in copy number variations between groups; **(B)** Differential expression of PSCA between unpaired tumour and normal tissue samples; **(C)** Differential expression of PSCA between paired tumour and normal tissue samples in combined TCGA and GTEx dataset; **(D)** Waterfall plot demonstrating the mutation frequency of PSCA in TCGA-LUAD dataset; **I** Domain information of single-nucleotide variants in TCGA-LUAD dataset; **(F)** Heat map of the correlation among six tumour stemness indices based on mRNA expression and methylation.

(S1F Fig), differentially methylated probe-based stemness index (S1C Fig) and enhancer element/DNA methylation-based stemness index (S1D Fig).

## Gene mutation analysis in pan-cancer in TCGA dataset

The cBioPortal database was used to evaluate the mutation frequency of PSCA based on pan-cancer data extracted from TCGA (Fig 2A). Amplification of the PSCA gene accounted for the majority of cancer types. The general mutation counts of PSCA in various cancer types in the cBioPortal database were visualised on bubble plots (Fig 2B and 2C). No mutation samples accounted for a large proportion, but amplification genotypes accounted for most of them, and the results were consistent. At the same time, it indicates that the Diploid in the sample is mostly. Subsequently, PSCA mutations identified in different cancer types were mapped to protein domains (Fig 2D), including mutations at ENST00000301258, ENST00000575167, ENST00000513264 and ENST00000571412.

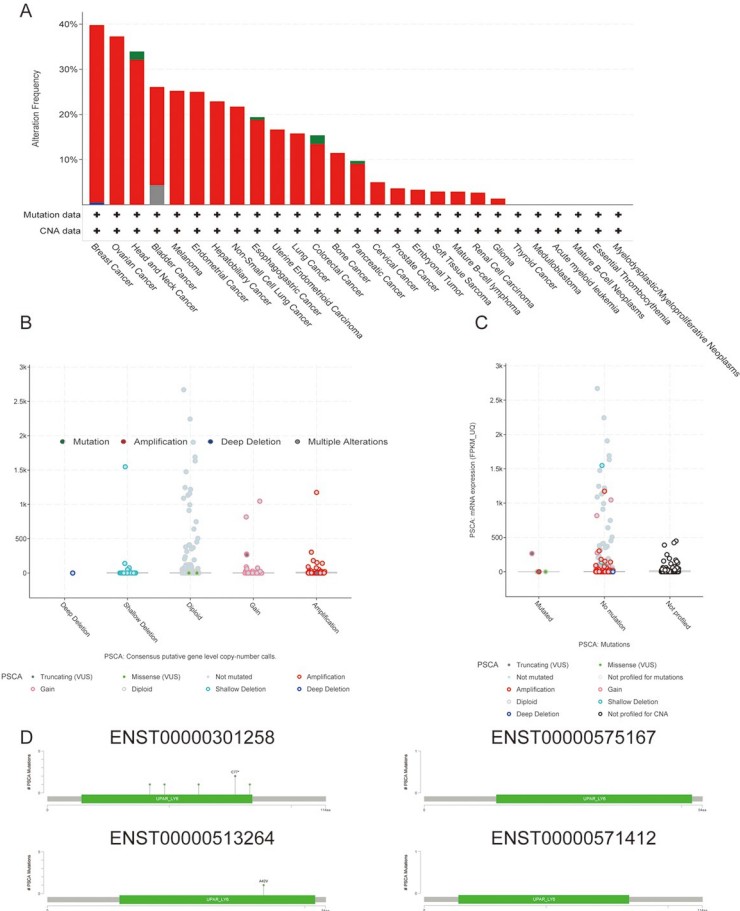

**Fig 2. Mutation pattern of PSCA in pan-cancer. (A)** The mutation frequency of PSCA was evaluated based on pan-cancer data from TCGA using the cBioPortal database (green represents mutation, red represents deep amplification, blue represents deletion and grey represents multiple alterations); **(B–C)** General mutation counts of PSCA in various cancer types in the cBioPortal database; **(D)** Mutation map of PSCA in protein domains of different cancer types.

## Effects of PSCA on genes related to RNA modification (m1A, m5C and m6A)

The correlation between the expression of PSCA and genes related to RNA modification (m1A, m5C and m6A), including NSUN5, DNMT3A, DNMT1, TRDMT1, HNRNPC and IGF2BP1, was significantly positive in CHOL, UCS, PAAD, ACC, LIHC, GBM, OV and SKCM and significantly negative in STAD, STES, KICH, KIPAN and KIRC (S2A Fig). Furthermore, the relationship between PSCA expression and each methylation site was examined. PSCA expression was strongly correlated with methylation at multiple sites, such as cg18270343 (r = 0.171, p-value < 0.001), cg10596483 (r = −0.157, p-value < 0.001), cg18270343 (r = −0.140, p-value = 0.007) and cg10596483 (r = −0.157, p-value < 0.001) (S2B–S2E Fig).

## Differential analysis of pathological features in pan-cancer

Histograms were generated to demonstrate specific molecular subtypes (S3A Fig), tumour stages (S3B Fig), OS rates (S3C Fig), immune cell subtypes (S3D Fig), grades (S3E Fig) and mutation difference between responders and non-responders (S3F Fig) in different cancer

types. BRCA showed significant differences in tumour stage, grade and other malignant characteristics. The genome-wide correlation of PSCA with other clinicopathological features in LUAD, GBM, STAD, OV, LUSC and BRCA is demonstrated in S4 Fig.

## Prognostic potential of PSCA in pan-cancer

A Cox proportional hazards regression model was established using the coxph function to evaluate the relationship between PSCA expression and prognosis in each tumour type. The log-rank test was used to assess the prognostic significance of PSCA. BRCA (N = 1044, p = 0.01, HR = 1.07 [1.02, 1.14]), LUAD (N = 490, p = 0.03, HR = 1.05 [1.00, 1.09]), GBM (N = 144, p = 0.05, HR = 1.07 [1.00, 1.15]), KIRC (N = 515, p = 0.03, HR = 1.07 [1.01, 1.13]), READ (N = 90, p = 0.02, HR = 1.27 [1.06, 1.53]), PAAD (N = 172, p = 9.0e-5, HR = 1.13 [1.06, 1.20]), high expression in 6 tumour types, namely, GBMLGG (N = 619, p = 3.0e-3, HR = 0.93 [0.89, 0.98]), CESC (N = 273, p = 0.03, HR = 0.92 [0.85, 0.99]), HNSC (N = 509, p = 0.04, HR = 0.94 [0.89, 1.00]), SKCM (N = 444, p = 0.03, HR = 0.95 [0.91, 1.00]), SKCM-M (N = 347, p = 0.04, HR = 0.95 [0.90, 1.00]), DLBC (N = 44, p = 9.8e-5, HR = 0.58 [0.42, 0.79]), low expression and poor prognosis (Table 1). A forest plot was generated to evaluate the hazard ratio and significance of PSCA in pan-cancer (Fig 3A). Survival curves demonstrated that low PSCA expression was associated with a poor prognosis in GBM, SKCM, CESC, HNSC and BRCA (Fig 3B–3F), whereas high PSCA expression was associated with a poor expression in KIRC, PAAD and READ (Fig 3G–3I).

The impact of PSCA on prognosis was assessed in multiple GEO datasets. PSCA was significantly associated with prognosis in LUAD, ovarian cancer and colorectal cancer. In particular, high PSCA expression was associated with poor OS in LUAegbcob-00182-UM dataset: cox. P = 0.004, HR = 1.44 [1.13–1.84]) (S5A Fig), whereas low PSCA expression was associated with poor OS in ovarian cancer (GSE17260 dataset: cox.P = 0.02, HR = 0.69 [0.51–0.94]; GSE8841 dataset: cox.P = 0.03, HR = 0.33 [0.12–0.89]) (S5C and S5D Fig). Low PSCA expression was associated with poor DFS in eye cancer (GSE22138 dataset: cox.P = 0.02, HR = 5.89 [1.35–25.66]) (S5F Fig). In addition, low PSCA expression was associated with poor OS (GSE17536 dataset: cox.P = 0.02, HR = 0.07 [0.01–0.67]) (S5B Fig) and DFS (GSE14333 dataset: cox.P = 0.001, HR = 1.34 [1.13–1.60]) (S5E Fig) in colorectal cancer. S1 Table shows the results of survival analysis in multiple datasets.

## Subgroups of clinicopathological characteristics in pan-cancer

Site-deletion mutations in PSCA were predominant in LUAD (S6A and S6B Fig). in CESC, HNSC, PSCA was significantly associated with the difference in T stage (S6C Fig), in GBMLGG, BRCA, In SKCM, PSCA was highly correlated with the difference in tumour stage (S6D Fig); in HNSC, SKCM, PSCA was highly correlated with the difference in N stage (S6E Fig); in pan-cancer, the difference between PSCA and tumour grade was not significant (S6F Fig), in COAD, THYM, READ, and BLCA, PSCA was significantly associated with differences in M staging (S6G Fig).

## Correlation analysis of immune checkpoints in pan-cancer

The expression of PSCA and 150 markers of five immune pathways (chemokines, receptors, MHC molecules, immunosuppressors and immunostimulators) was examined, and the correlation between PSCA and the marker genes was visualised on a heat map (Fig 4A). The expression of MHC molecules, receptors, immunosuppressors, immunostimulators and chemokine-related genes was compared between high- and low-PSCA-expression groups in

**Table 1. Pan-cancer prognostic analysis of PSCA.**

| CancerCode | p-value | HR (95%CI) |
| --- | --- | --- |
| TCGA-PAAD(N = 172) | 0.00 | 1.13(1.06,1.20) |
| TCGA-BRCA(N = 1044) | 0.01 | 1.07(1.02,1.14) |
| TCGA-READ(N = 90) | 0.02 | 1.27(1.06,1.53) |
| TCGA-LUAD(N = 490) | 0.03 | 1.05(1.00,1.09) |
| TCGA-KIRC(N = 515) | 0.03 | 1.07(1.01,1.13) |
| TCGA-GBM(N = 144) | 0.05 | 1.07(1.00,1.15) |
| TCGA-THYM(N = 117) | 0.05 | 1.37(1.01,1.86) |
| TCGA-MESO(N = 84) | 0.06 | 1.12(1.00,1.26) |
| TCGA-LIHC(N = 341) | 0.09 | 1.05(0.99,1.11) |
| TCGA-LAML(N = 209) | 0.11 | 1.05(0.99,1.11) |
| TCGA-PCPG(N = 170) | 0.17 | 1.35(0.88,2.07) |
| TCGA-SARC(N = 254) | 0.20 | 1.05(0.97,1.14) |
| TCGA-STAD(N = 372) | 0.30 | 1.02(0.98,1.07) |
| TARGET-NB(N = 151) | 0.30 | 1.05(0.96,1.16) |
| TCGA-LUSC(N = 468) | 0.34 | 1.03(0.97,1.09) |
| TCGA-COADREAD(N = 368) | 0.40 | 1.03(0.96,1.10) |
| TCGA-STES(N = 547) | 0.49 | 1.01(0.98,1.05) |
| TCGA-ACC(N = 77) | 0.51 | 1.04(0.93,1.17) |
| TCGA-KIRP(N = 276) | 0.55 | 1.03(0.93,1.14) |
| TARGET-WT(N = 80) | 0.67 | 1.02(0.92,1.14) |
| TARGET-ALL-R(N = 99) | 0.92 | 1.00(0.94,1.08) |
| TCGA-DLBC(N = 44) | 0.00 | 0.58(0.42,0.79) |
| TCGA-GBMLGG(N = 619) | 0.00 | 0.93(0.89,0.98) |
| TCGA-CESC(N = 273) | 0.03 | 0.92(0.85,0.99) |
| TCGA-SKCM(N = 444) | 0.03 | 0.95(0.91,1.00) |
| TCGA-HNSC(N = 509) | 0.04 | 0.94(0.89,1.00) |
| TCGA-SKCM-M(N = 347) | 0.04 | 0.95(0.90,1.00) |
| TCGA-BLCA(N = 398) | 0.09 | 0.97(0.94,1.00) |
| TCGA-LGG(N = 474) | 0.22 | 0.95(0.88,1.03) |
| TCGA-UCEC(N = 166) | 0.26 | 0.92(0.81,1.06) |
| TARGET-LAML(N = 142) | 0.29 | 0.97(0.91,1.03) |
| TCGA-SKCM-P(N = 97) | 0.40 | 0.92(0.75,1.12) |
| TCGA-KICH(N = 64) | 0.57 | 0.95(0.81,1.12) |
| TCGA-ESCA(N = 175) | 0.62 | 0.98(0.92,1.05) |
| TCGA-TGCT(N = 128) | 0.65 | 0.89(0.55,1.46) |
| TCGA-OV(N = 407) | 0.75 | 0.99(0.95,1.04) |
| TCGA-UCS(N = 55) | 0.75 | 0.98(0.84,1.13) |
| TCGA-UVM(N = 74) | 0.80 | 0.97(0.74,1.26) |
| TCGA-THCA(N = 501) | 0.82 | 0.98(0.84,1.15) |
| TCGA-COAD(N = 278) | 0.86 | 0.99(0.92,1.07) |
| TCGA-KIPAN(N = 855) | 0.89 | 1.00(0.96,1.04) |
| TARGET-ALL(N = 86) | 0.89 | 0.99(0.92,1.08) |
| TCGA-PRAD(N = 492) | 0.91 | 0.99(0.78,1.25) |
| TCGA-CHOL(N = 33) | 0.97 | 1.00(0.83,1.20) |

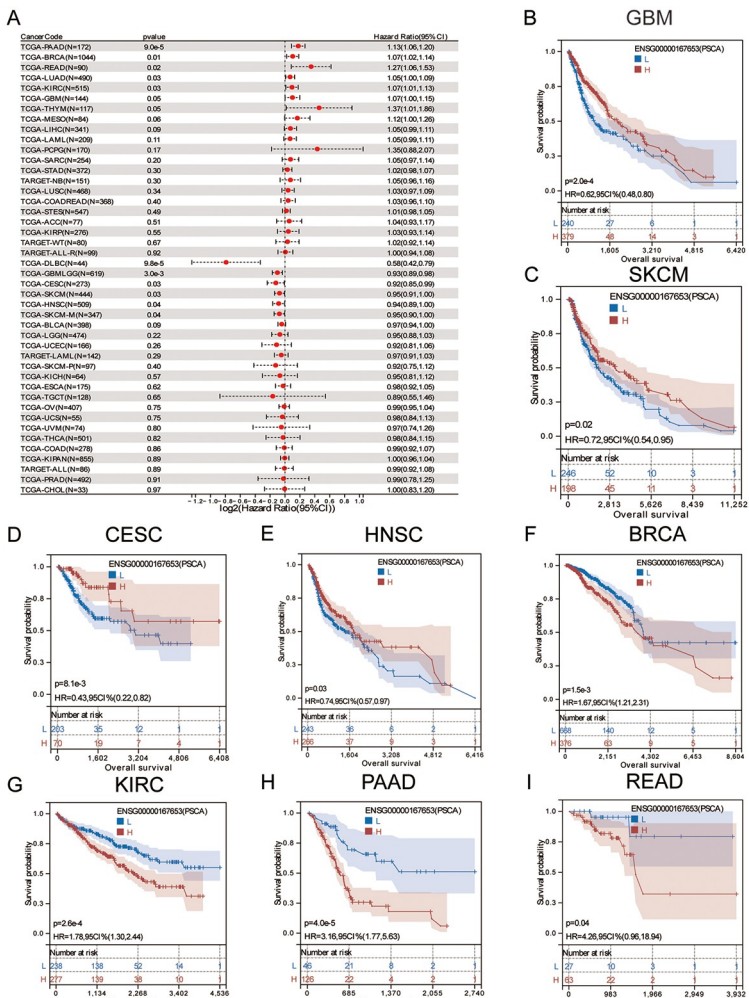

**Fig 3. Prognostic analysis of PSCA in TCGA, TARGET and GTEx databases in pan-cancer. (A)** Log-rank test was used to obtain a significant forest plot of the risk of pan-cancer overall survival prognosis to quantify the hazard ratio and significance of PSCA in pan-cancer; **(B–F)** Low PSCA expression was associated with a poor prognosis in GBM, SKCM, CESC, HNSC and BRCA; **(G–I)** High PSCA expression was associated with a poor prognosis in KIRC, PAAD and READ.

LUAD dataset using TISIDB, and the results were visualised on a heat map. Differences in the expression of immunostimulators between the two groups were most predominant (Fig 4B–4F).

S7 Fig shows the gene expression valueICNA and methylation levels of lymphocytes. The expression of immunosuppressors, immunostimulators and MHC molecules was evaluated based on PSCA expression in each sample.

## Univariate and multivariate regression analyses of the prognostic potential of PSCA in LUAD

Univariate and multivariate regression analyses were performed to identify independent factors influencing the prognosis and survival of patients with LUAD. A forest plot was generated to demonstrate PSCA expression and clinicopathological data, including pathological stages,

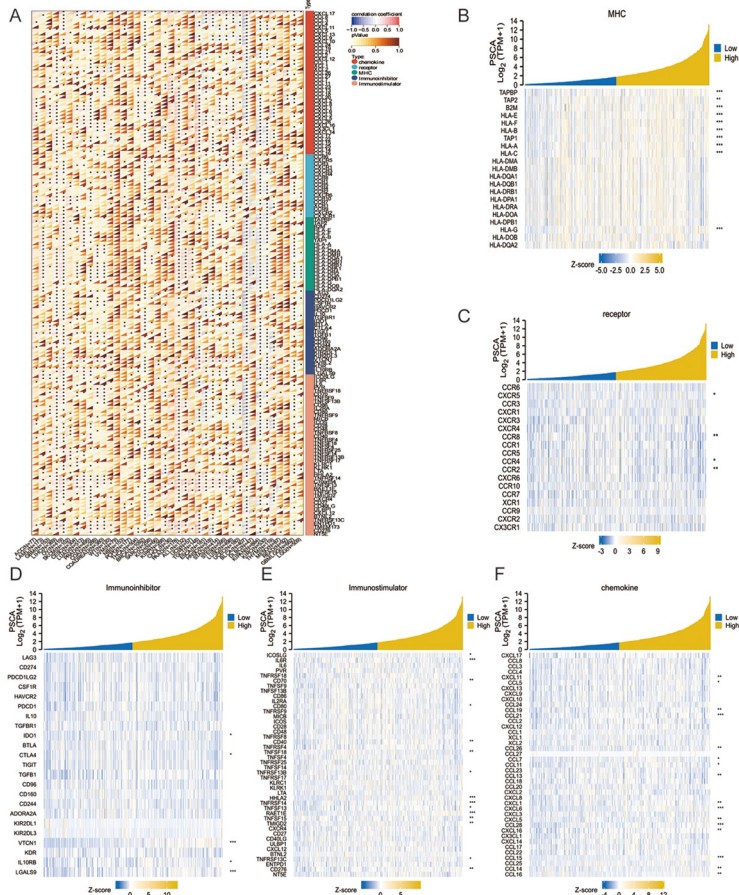

**Fig 4. PSCA affects immune regulation in tumours. (A)** Expression of PSCA and five types of marker genes in each sample; **(B–F)** Heat map demonstrating the differential expression of MHC molecules, receptors, immunosuppressors, immunostimulators and chemokine-related genes between the high- and low-PSCA-expression groups in LUAD.

TNM stages and outcomes (S8A and S8B Fig and S2 Table). Samples were divided into subgroups based on T stage, N stage, M stage, primary treatment outcomes and pathologic stages. The results of risk regression analysis were visualised on forest plots. High expression of PSCA, N stage and primary treatment outcomes were identified as independent prognostic factors affecting survival in LUAD. A nomogram integrating PSCA and clinical characteristics was generated to quantify the impact of risk factors on the prognosis of LUAD, and calibration curves were plotted to validate the predictive accuracy of the nomogram (S8C and S8E Fig). K-M curves indicated that high PSCA expression was significantly associated with poorer OS in LUAD (S8D Fig).

A logistic regression model was established to evaluate the relationship between PSCA expression and clinicopathological features such as N stage and residual tumour (S3 Table). Consistent with the Cox model, the logistic regression model demonstrated a strong correlation between the two clinicopathological features (N1, N2, N3 and N0) and PSCA expression.

## Effects of PSCA expression on prognosis in LUAD

To examine the prognostic significance of PSCA in LUAD, survival analysis was performed in subgroups based on different clinicopathological characteristics. High expression of PSCA was

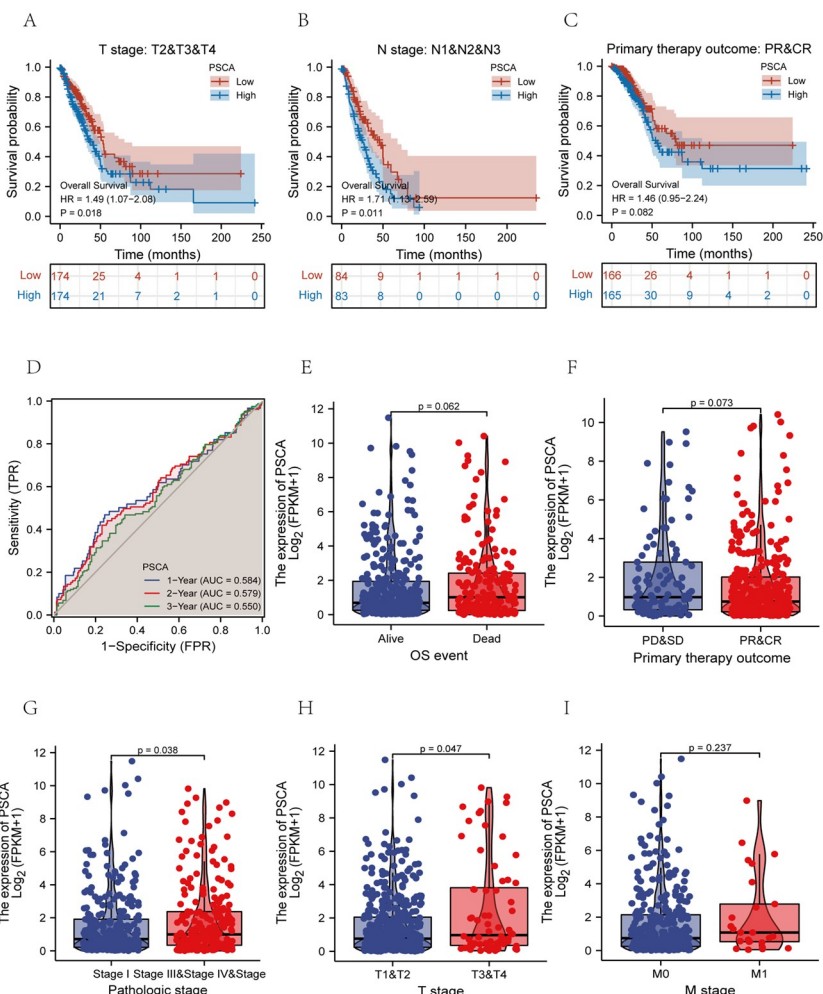

**Fig 5. Prognostic significance of PSCA in LUAD. (A–C)** Survival analysis in different clinicopathological subgroups based on T stage (T2, T3 and T4 groups), N stage (N1, N2 and N3 groups) and primary treatment outcomes (PR and CR groups); **(D)** Time-dependent ROC curves demonstrating survival at 1, 3 and 5 years in patients with LUAD; **(E–I)** Discrimination ability of PSCA for unfavourable clinicopathological characteristics in LUAD, including T stage, N stage, M stage, treatment outcomes and pathological stage.

found to be significantly associated with poorer survival in subgroups based on T stage, N stage and primary treatment outcomes (PR and CR groups) (Fig 5A–5C). Based on the survival status and OS time of patients with LUAD, the 1-, 3- and 5-year survival probabilities were assessed using time-dependent ROC curves [26]. The results showed that AUC values at 1, 2 and 3 years were 0.584, 0.579 and 0.550, respectively (Fig 5D).

To examine the ability of PSCA to distinguish unfavourable clinicopathological characteristics in LUAD, PSCA expression was compared between/among subgroups based on T stage, N stage, M stage, treatment outcomes and pathological stage (Fig 5E–5I) and clinical variables Subgroup ROC curves (S9 Fig).

In LUAD, high PSCA expression was significantly associated with advanced pathological stage and grade and poor OS rates (Fig 5E). PD and SD in primary therapy outcome subgroup, pathologic stage III and IV subgroup, T2 and T3 and T4 subgroup in T stage, and M1 subgroup in M stage were significantly overexpressed in poor prognosis subgroup (Fig 5F–5I).

In addition to verifying the differential efficacy of PSCA for LUAD (AUC = 0.549, CI = 0.497−0.600) (S9A Fig), we examined the efficacy of PSCA in distinguishing the above-mentioned clinicopathological features. PSCA had better discrimination power for TNM stage, residual tumour and primary treatment outcomes (S9B–S9F Fig), with the identification of residual tumour being particularly significant (AUC = 0.609, CI = 0.469−0.749). These results suggest that high PSCA expression is closely related to the poor prognosis of LUAD.

## ESTIMATE algorithm base on PSCA expression infers tumour purity in LUAD

The ESTIMATE algorithm was used to investigate the immune infiltration landscape of LUAD. The top 3 important correlations between PSCA and immune cells were visualised on scatter plots (Fig 6A–6C). PSCA expression was negatively correlated with stromal, immune

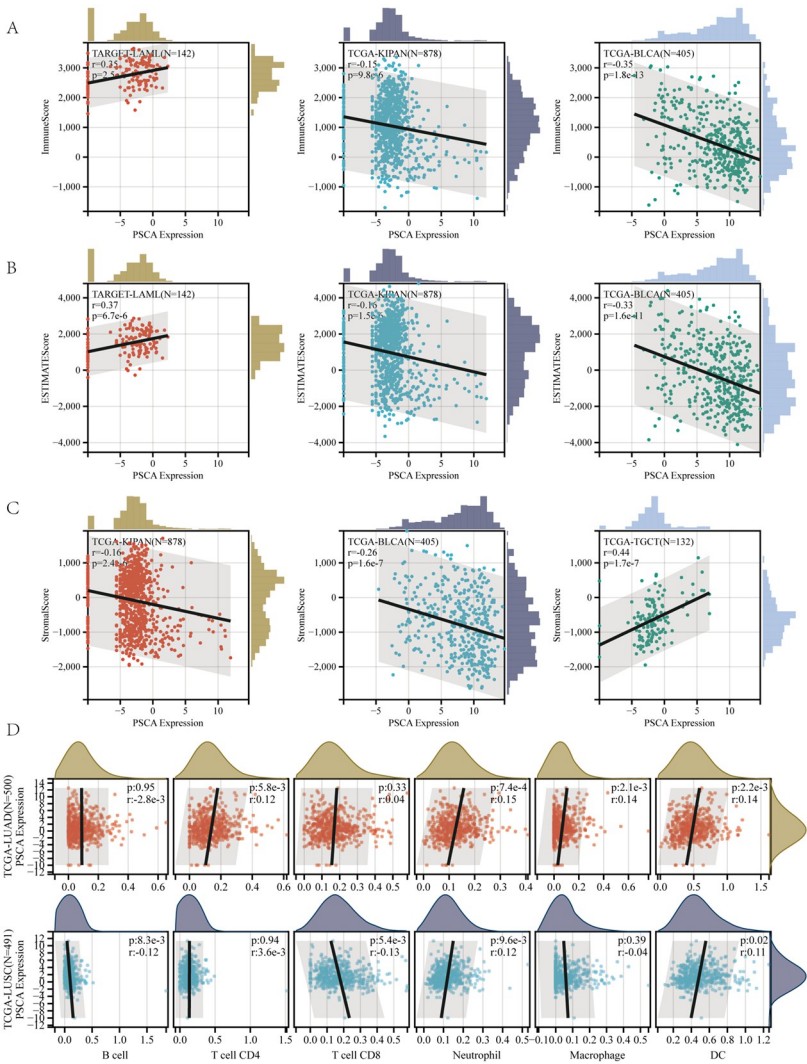

**Fig 6. Stromal, immune and ESTIMATE scores were calculated for each patient in each tumour dataset based on PSCA expression. (A–C)** Scatter plots demonstrating the three most significant correlations; **(D)** Scatter plots demonstrating the correlation between PSCA expression and the abundance of B cells, CD4 T cells, CD8 T cells, neutrophils, macrophages and DC in LUAD and LUSC as assessed using the TIMER algorithm.

**Table 2. TIMER counter deconvolution to analyze the infiltration of 6 major immune cells in LUAD & LUSC.**

| | LUAD | | LUSC | |
|---|---|---|---|---|
| | R | P | R | P |
| B_cells_naive | 0.02 | 0.66 | 0.04 | 0.04 |
| B_cells_memory | 0.09 | 0.04 | -0.01 | -0.01 |
| Plasma_cells | -0.05 | 0.29 | 0.06 | 0.06 |
| T_cells_CD8 | 0.03 | 0.5 | -0.17 | -0.17 |
| T_cells_CD4_naive | -0.02 | 0.58 | 0 | 0 |
| T_cells_CD4_memory_resting | -0.12 | 0.01 | -0.08 | -0.08 |
| T_cells_CD4_memory_activated | -0.09 | 0.05 | -0.09 | -0.09 |
| T_cells_follicular_helper | 0.11 | 0.01 | -0.01 | -0.01 |
| T_cells_regulatory_(Tregs) | 0.01 | 0.75 | 0.08 | 0.08 |
| T_cells_gamma_delta | 0.06 | 0.18 | 0.02 | 0.02 |
| NK_cells_resting | 0.05 | 0.22 | 0.05 | 0.05 |
| NK_cells_activated | 0 | 0.98 | 0 | 0 |
| Monocytes | -0.04 | 0.37 | 0.12 | 0.12 |
| Macrophages_M0 | 0.14 | 0 | 0.01 | 0.01 |
| Macrophages_M1 | 0.06 | 0.21 | -0.14 | -0.14 |
| Macrophages_M2 | -0.02 | 0.65 | 0.01 | 0.01 |
| Dendritic_cells_resting | -0.18 | 0 | -0.02 | -0.02 |
| Dendritic_cells_activated | 0.01 | 0.88 | 0.17 | 0.17 |
| Mast_cells_resting | -0.03 | 0.48 | 0.1 | 0.1 |
| Mast_cells_activated | 0 | 0.91 | 0.02 | 0.02 |
| Eosinophils | 0.1 | 0.02 | -0.07 | -0.07 |
| Neutrophils | 0.01 | 0.77 | 0.13 | 0.13 |

and estimate scores in KIPAN and BLCA, indicating that samples with low immune infiltration levels had high PSCA expression. These results are consistent with those described in previous sections, indicating that patients with high PSCA expression have a poor prognosis. The TIMER algorithm was used to assess the correlation between immune cell infiltration and PSCA expression in LUAD and LUSC, and the results were visualised on a scatter plot (Fig 6D and Table 2). PSCA expression was significantly positively correlated with the abundance of Th cells, neutrophils, macrophages and DCs in LUAD. However, the correlation between PSCA expression and immune cell infiltration was not significant in LUSC.

## Immune checkpoint gene expression and immune infiltration analyses

We identified key immune checkpoint genes reported in the literature and found that the genes were significantly associated with PSCA (Fig 7A). The abundance of NK cells, CD56dim NK cells, neutrophils, mast cells, Treg cells, Th2 cells, Th1 cells, Tgd cells, Tcm cells, macrophages, aDCs and eosinophils was closely associated with the expression of PSCA in LUAD (Fig 7B). Similarly, PSCA expression was significantly correlated with the degree of infiltration of immune cells in other cancer types (Fig 7C). The correlation between the expression of PSCA and abundance of tumour-infiltrating immune cells was visualised on a lollipop chart (Fig 7D). PSCA expression was negatively correlated with the abundance of Tem and T helper cells (Fig 7F and 7G) and significantly positively correlated with that of Tgd cells, neutrophils and CD56bright NK cells (Fig 7H–7J). To characterise molecular mechanisms in the immune microenvironment, we analysed the relationship between PSCA and six immune subtypes,

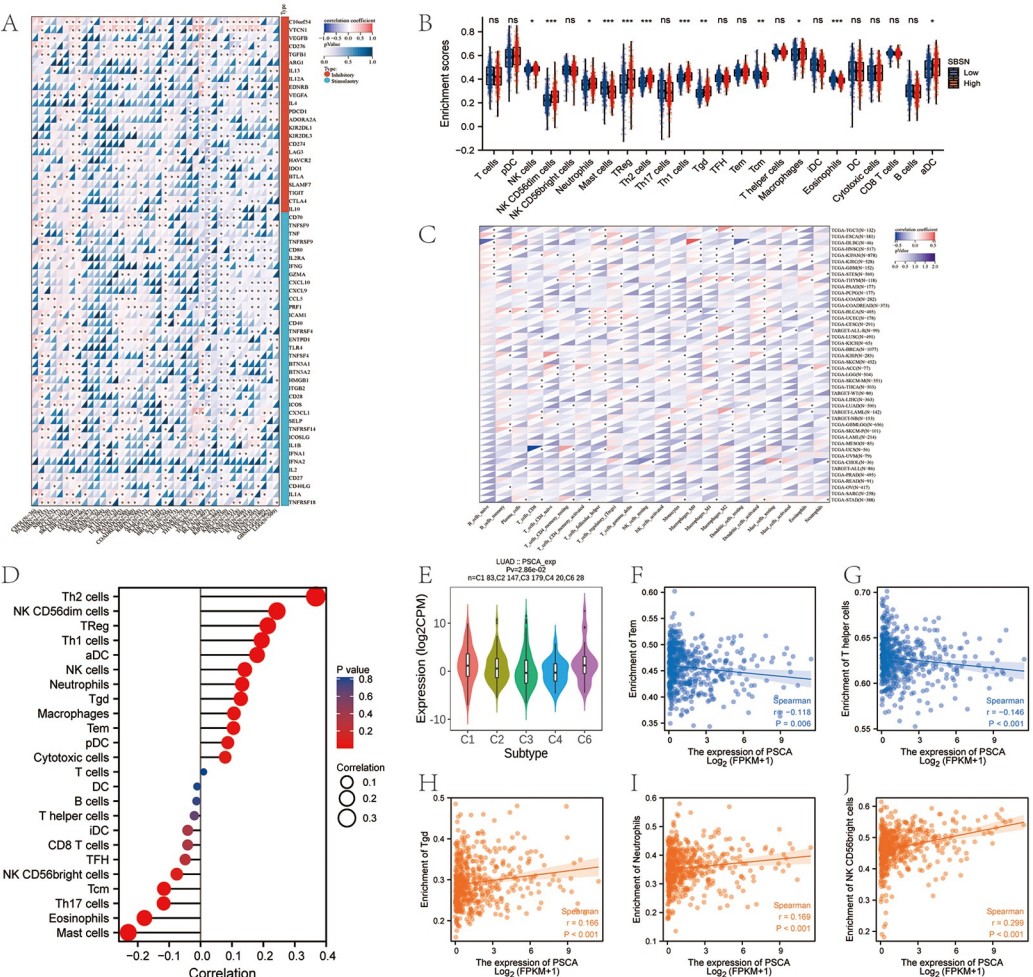

**Fig 7. Analysis of immune checkpoint gene expression and immune cell infiltration.** (A) Heat map demonstrating the correlation between PSCA and immune checkpoint genes; (B) The CIBERSORT algorithm was used to estimate the proportion of 22 immune cell types. Box plot demonstrates differences in the abundance of immune cells between the high- and low-PSCA-expression groups; (C) 22 types of immune cell infiltration in PSCA cancer (*, P < 0.05; **, P < 0.01; ***, P < 0.001); (D) Lollipop chart demonstrating the correlation between PSCA expression and immune cell infiltration in ID; (E) Relationship between PSCA expression and immune and molecular subtypes in LUAD; (F–G) Scatter plot demonstrating a negative correlation between PSCA expression and the abundance of Tem and T helper cells; (H–J) Scatter plot demonstrating a significantly positive correlation between PSCA expression and the abundance of Tgd cells, neutrophils and CD56bright NK cells.

including C1 (wound healing), C2 (IFN-gamma dominant), C3 (inflammatory), C4 (lymphocyte depletion), C5 (immunologically quiet) and C6 (TGF-b dominant) subtypes (Fig 7E). PSCA was found to be enriched in the C1 and C6 subtypes, suggesting that PSCA affects the immune microenvironment of LUAD through these two pathways.

## GO and KEGG enrichment analyses

GO [27] functional annotation and KEGG [28] pathway enrichment analyses (Fig 8 and Table 3) indicated that PSCA is closely related to substance metabolism, enzyme inhibitor activity, cell proliferation, structural components of the cytoskeleton and dynein complex binding. The results of GO and KEGG analyses were visualised on histograms and bubble

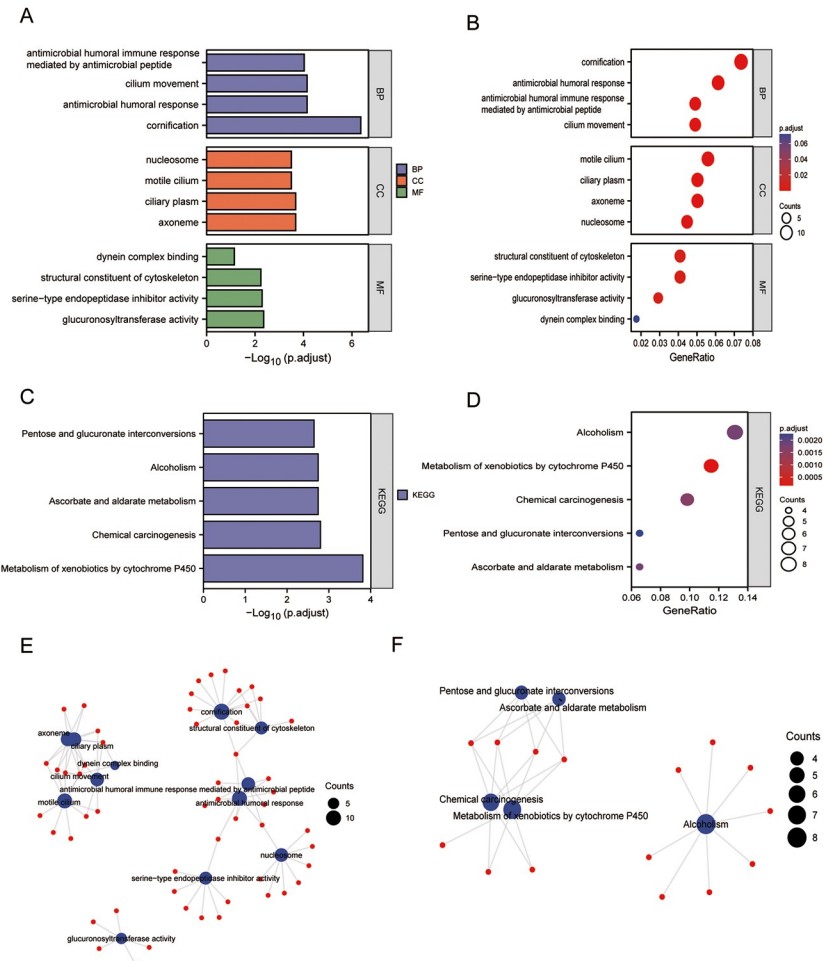

**Fig 8. GO and KEGG enrichment analyses of genes co-expressed with PSCA. (A–B)** Histogram and bubble plot of GO analysis; **(C–D)** Histogram and bubble plot of KEGG analysis; **(E–F)** Network map of GO and KEGG enrichment analyses.

plots (Fig 8A, 8B and 8E). In particular, GO analysis revealed that PSCA was enriched in biological processes such as cornification, antimicrobial humoral response, cilium movement, antimicrobial humoral immune response mediated by antimicrobial peptides and keratinocyte differentiation; molecular functions such as glucuronosyltransferase activity, serine-type endopeptidase inhibitor activity, structural constituent of the cytoskeleton, dynein complex binding and endopeptidase inhibitor activity and cellular components such as the axoneme, ciliary plasm, motile cilium, nucleosome and DNA packaging complex.

KEGG analysis revealed that genes co-expressed with PSCA were mostly enriched in pathways associated with the metabolism of xenobiotics by cytochrome P450, chemical carcinogenesis, ascorbate and aldarate metabolism, alcoholism and pentose and glucuronate interconversion (Fig 8C, 8D and 8F).

## GSEA based on differential expression of PSCA

Gene set enrichment analysis (GSEA) was performed using the 'c2.cp.kegg.v7.4.entrez.gmt' gene set from the Molecular Signatures Database (MSigDB) (S4 Table). Analysis of data

**Table 3. GO&KEGG enrichment analysis of PSCA co-expressed genes.**

| ONTOLOGY | ID | Description | p.adjust | qvalue |
|---|---|---|---|---|
| BP | GO:0070268 | cornification | 4.31e-07 | 4.12e-07 |
| BP | GO:0019730 | antimicrobial humoral response | 7.14e-05 | 6.82e-05 |
| BP | GO:0003341 | cilium movement | 7.14e-05 | 6.82e-05 |
| BP | GO:0061844 | antimicrobial humoral immune response mediated by antimicrobial peptide | 9.38e-05 | 8.96e-05 |
| BP | GO:0030216 | keratinocyte differentiation | 1.55e-04 | 1.48e-04 |
| CC | GO:0005930 | axoneme | 2.11e-04 | 1.98e-04 |
| CC | GO:0097014 | ciliary plasm | 2.11e-04 | 1.98e-04 |
| CC | GO:0031514 | motile cilium | 3.19e-04 | 2.99e-04 |
| CC | GO:0000786 | nucleosome | 3.19e-04 | 2.99e-04 |
| CC | GO:0044815 | DNA packaging complex | 4.35e-04 | 4.08e-04 |
| MF | GO:0015020 | glucuronosyltransferase activity | 0.004 | 0.004 |
| MF | GO:0004867 | serine-type endopeptidase inhibitor activity | 0.005 | 0.005 |
| MF | GO:0005200 | structural constituent of cytoskeleton | 0.006 | 0.005 |
| MF | GO:0070840 | dynein complex binding | 0.072 | 0.065 |
| MF | GO:0004866 | endopeptidase inhibitor activity | 0.072 | 0.065 |
| KEGG | hsa00980 | Metabolism of xenobiotics by cytochrome P450 | 1.54e-04 | 1.30e-04 |
| KEGG | hsa05204 | Chemical carcinogenesis | 0.002 | 0.001 |
| KEGG | hsa00053 | Ascorbate and aldarate metabolism | 0.002 | 0.002 |
| KEGG | hsa05034 | Alcoholism | 0.002 | 0.002 |
| KEGG | hsa00040 | Pentose and glucuronate interconversions | 0.002 | 0.002 |

extracted from the REACTOME database revealed that DNA methylation, DNA damage telomere stress-induced senescence, transcriptional regulation by small RNAs, RNA polymerase transcription, DNA double-strand break repair and G2–M DNA damage checkpoint pathway were significantly enriched in the low-PSCA-expression group (S10A–S10F Fig). The CD22-mediated BCR regulation pathway was significantly enriched in the high-PSCA-expression group (S10J Fig). The P53 signalling pathway in the KEGG database (S10K Fig), the small cell lung cancer-related pathway in the WP database (S10L Fig), the IL-4/-2 and TF pathways in the PID database (S10H and S10I Fig) and the MHC pathway in the Biocarta database (S10G Fig) were significantly enriched in the PSCA-high-expression group.

## PSCA promotes cell migration and invasion in lung cancer

Western blotting and qRT-PCR showed that the expression of PSCA was higher in lung cancer tissues than in para-cancerous tissues (Fig 9A and 9B). Subsequently, the expression of PSCA was examined in four lung cancer cell lines. The expression of PSCA was higher in H1299 and A549 cells (Fig 9C and 9D). Three siRNAs were used to knock down PSCA in H1299 and A549 cells, with si-PSCA-1 having the most significant effect (Fig 9E and 9F). PSCA knockdown attenuated the migratory and invasive capabilities of H1299 and A549 cells (Fig 9G and 9H) and enhanced apoptosis (Fig 9I and 9K). In addition, PSCA knockdown prompted lung cancer cells to exit the cell cycle (Fig 9J and 9L).

## Discussion

PSCA is associated with disease progression, promotion of angiogenesis, invasion, metastasis and immune evasion in cancer [4,29]. However, its expression pattern as well as diagnostic and prognostic potential have not been thoroughly analysed from a pan-cancer perspective

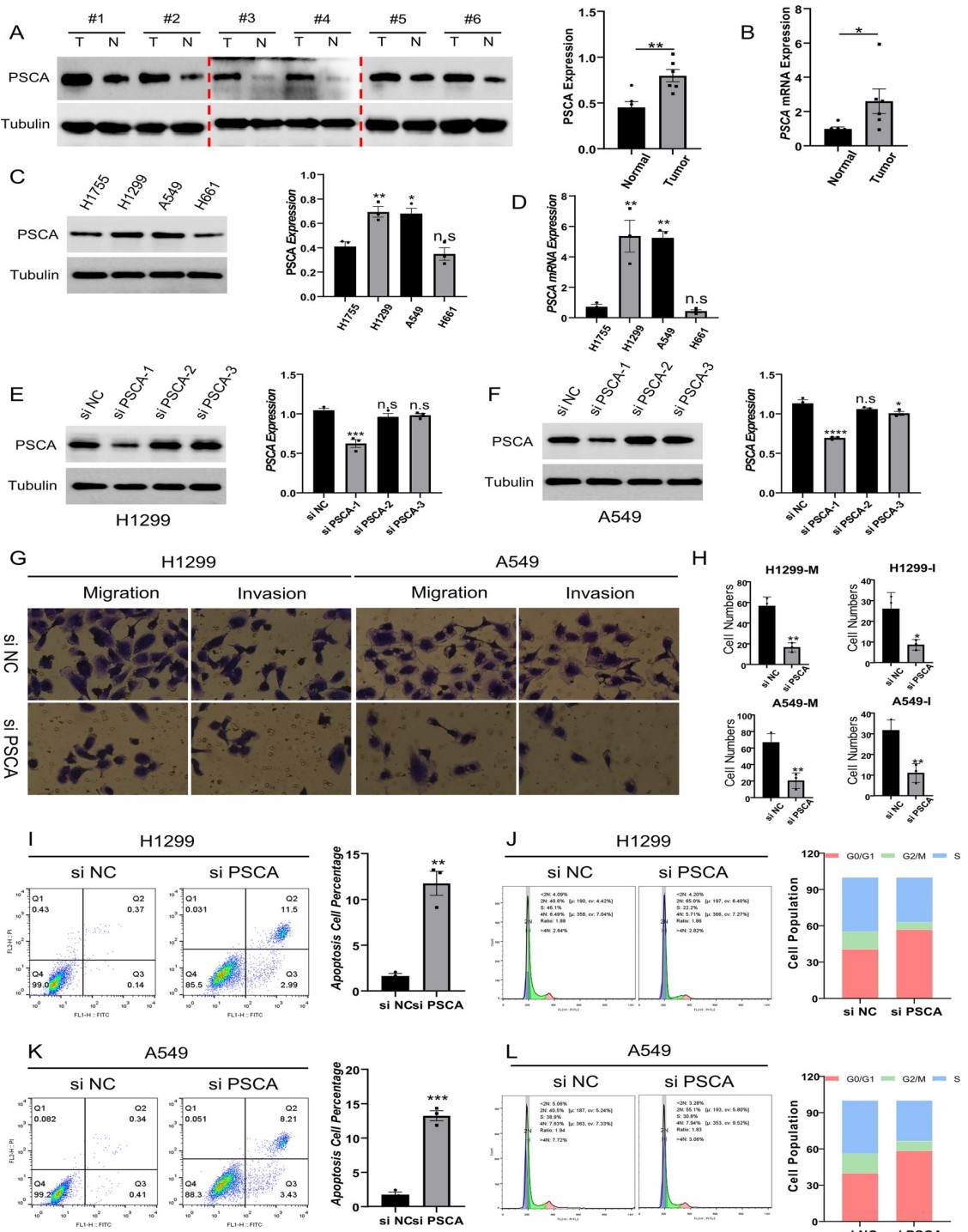

**Fig 9. PSCA expression is upregulated in tumour tissues and promotes tumour cell migration, invasion and proliferation. (A–B)** Western blotting and qRT-PCR were performed to evaluate the mRNA and protein expression of PSCA in six pairs of tumour tissues (T) and adjacent normal tissues (N); **(C–D)** Protein and mRNA expression of PSCA in four lung cancer cell lines; **(E–F)** Quantitative analysis of the expression of PSCA in control and PSCA-knockdown H1299 and A549 cells; **(G)** Transwell assay was performed to assess cell migratory and invasion in the negative control and PSCA-knockdown groups; **(H)** Quantitative analysis of migrating and invading tumour cells; **(I–L)** Flow cytometric detection of apoptotic cells and cell cycle in control and PSCA-knockdown cells (n.s, p-value>0.05; *, p-value<0.05; **, p-value<0.01; ***, p-value<0.001; ****, p-val = e<0.0001).

[8]. In this study, we examined the expression and prognostic value of PSCA in pan-cancer using the expression data of 33 cancer types from TCGA and GTEx databases. PSCA expression was upregulated in UCEC, BRCA and ESCA and downregulated in GBM, KIRP, PRAD, STAD, HNSC, KIRC and READ. Multi-omic analysis of LUAD data revealed that multiple differentially expressed genes, including PSCA, were associated with an increased risk of disease recurrence. The frequency of deep amplifications in PSCA was high in LUAD samples, which supported the results of the multi-omic analysis. PSCA expression was positively correlated with ENHss, EREG-METHss and DNAss. In addition, the correlation between the expression of PSCA and genes related to RNA modification (m1A, m5C and m6A), including NSUN5, DNMT3A, DNMT1, TRDMT1, HNRNPC and IGF2BP1, was positive in CHOL, UCS, PAAD, ACC, LIHC, GBM, OV and SKCM but negative in STAD, STES, KICH, KIPAN and KIRC. In addition, PSCA expression was significantly associated with specific molecular subtypes, tumour stage, grade, OS and immune cell subtypes in pan-cancer. Altogether, these findings suggest that PSCA expression varies across different clinicopathological characteristics of tumours, such as pathological stages and grades.

Cox regression analysis suggested that high PSCA expression was associated with a high risk of BRCA, LUAD, GBM, KIRC, READ and PAAD, whereas low PSCA expression was associated with a high risk of CESC, HNSC, SKCM and DLBC. KM curves demonstrated that low PSCA expression was associated with a poor prognosis in GBM, SKCM, CESC, HNSC and BRCA, whereas high PSCA expression was associated with a poor prognosis in KIRC, PAAD and READ in TCGA datasets. In addition, high PSCA expression was associated with poor prognosis and survival in LUAD, ovarian cancer and colorectal cancer in GEO datasets. Subgroup analysis based on clinicopathological characteristics showed that PSCA was significantly correlated with tumour stage, grade and TNM stage in CESC, HNSC, PSCA, GBMLGG, BRCA, SKCM, COAD, THYM, READ and BLCA (S6G Fig).

To determine the immunological mechanisms through which PSCA affects cancer development and progression, we analysed the expression data of 150 marker genes for five immune pathways (chemokines, receptors, MHC molecules, immunosuppressors and immunostimulators) [13,14,30]. The results suggested that PSCA affects cancer development and progression through immunostimulatory factors [30].

The prognostic role of PSCA in LUAD was analysed based on pathological stage, TNM stage and other clinicopathological features. Univariate and multivariate regression analyses validated that N stage and primary treatment outcomes were independent factors affecting the prognosis and survival of patients with LUAD. Nomograms and calibration curves were plotted to quantify and verify the prognostic value of each clinical variable. The results suggested that high PSCA expression was significantly associated with poorer OS. Time-dependent ROC curves demonstrating the 1-, 3- and 5-year OS probabilities of patients with LUAD validated the ability of PSCA to distinguish poor prognostic clinicopathological subgroups. High PSCA expression was significantly associated with advanced pathological stages and grades, poor OS rates and worse prognosis.

To investigate the immune infiltration landscape of LUAD, the ESTIMATE algorithm was used to calculate tumour purity scores based on PSCA expression. Subsequently, the CIBERSORT and TIMER algorithms were used to evaluate the correlation between immune cell infiltration and PSCA expression. In tumour types such as LUAD, BLCA and KIRP, high PSCA expression was significantly negatively correlated with the abundance of various immune cell types, suggesting that patients with high PSCA expression had low immune cell infiltration, which is consistent with the overall poor prognosis. The differential expression of PSCA among the six immune subtypes suggested that the immune microenvironment of LUAD is influenced by mechanisms related to wound healing, IFN-γ dominance and TGF-β

dominance. Similarly, GO and KEGG enrichment analyses of genes co-expressed with PSCA and GSEA of PSCA showed that PSCA was significantly enriched in disease pathology- and substance metabolism-related pathways.

In conclusion, our study reveals the differential expression and prognostic value of PSCA in different tumour tissues, which provides assistance in the clinical diagnosis and assessment of tumours. Meanwhile, we found that PSCA is closely associated with the immune infiltration status of tumours, and this may make it a new target for immune intervention for the treatment of tumours.

## Conclusions

Elevated PSCA expression may affect the prognostic value of pan-cancer by changing the degree of immune infiltration. Especially in LUAD, high PSCA expression was associated with worse survival outcomes and lower immune cell infiltration. We performed a comprehensive assessment of PSCA, revealing its potential role as a patient prognostic indicator and its immunomodulatory role.

## Supporting information

**S1 Fig. Relationship between PSCA and six tumour stemness indices in LUAD. (A)** DNA methylation-based stemness score (DNAss); **(B)** RNA-based stemness score (RNAss); **(C)** Differential methylation probe-based stemness score (DMPss); **(D)** Enhancer element/DNA methylation-based stemness score (ENHss); **(E)** Epigenetically regulated RNA-based stemness score (EREG.EXPss); **(F)** Epigenetically regulated DNA methylation-based stemness score (EREG-METHss).
(DOCX)

**S2 Fig. Effects of PSCA on genes related to RNA modification (m1A, m5C and m6A). (A)** Correlation between the expression of PSCA and RNA modification-related genes in pan-cancer; **(B–E)** PSCA is closely related to methylation modifications at multiple sites, including cg18270343 (r = 0.171, P < 0.001), cg10596483 (r = −0.157, P < 0.001), cg18270343 (r = −−0.140, P = 0.007) and cg10596483 (r = −0.157, P < 0.001).
(DOCX)

**S3 Fig. Pan-cancer analysis of pathological features.** Histograms demonstrate the relationship between PSCA expression and specific molecular subtypes of human cancers **(A)**, stage **(B)**, overall survival **(C)**, immune cell subtypes **(D)**, grade **(E)** and mutation difference between responders and non-responders **(F)** in pan-cancer.
(DOCX)

**S4 Fig. Genome-wide correlation analysis.** Genome-wide correlation between PSCA and other signatures in LUAD **(A)**, GBM **(B)**, STAD **(C)**, OV **(D)**, LUSC **(E)** and BRCA **(F)**.
(DOCX)

**S5 Fig. Validation of prognostic significance of PSCA in pan-cancer. (A)** OS in jacob-00182-UM dataset (cox.P = 0.004, HR = 1.44 [1.13–1.84]); **(B)** OS in GSE17536 dataset (cox.P = 0.02, HR = 0.07 [0.01–0.67]); **(C)** OS in ovarian cancer dataset GSE17260 (cox.P = 0.02, HR = 0.69 [0.51–0.94]); **(D)** OS in GSE8841 dataset (cox.P = 0.03, HR = 0.33 [0.12–0.89]); **(E)** DFS in GSE14333 dataset (cox.P = 0.001, HR = 1.34 [1.13–1.60]); **(F)** DFS in the eye cancer dataset GSE22138 (cox.P = 0.02, HR = 5.89 [1.35–25.66]).
(DOCX)

**S6 Fig. Subgroup analysis based on clinicopathological characteristics in pan-cancer. (A–B)** Mutation pattern of PSCA in LUAD; **(C)** PSCA expression was significantly correlated with T stage in CESC and HNSC; **(D)** PSCA expression was significantly correlated with tumour stage in GBMLGG, BRCA and SKCM; **(E)** PSCA expression was significantly correlated with N stage in HNSC and SKCM; **(F)** The relationship between PSCA expression and tumour grade was not significant in pan-cancer; **(G)** PSCA expression was significantly correlated with M stage in COAD, THYM, READ and BLCA.
(DOCX)

**S7 Fig. PSCA affects immune regulation in tumours.** In each cancer sample, the gene expression values, CNA and methylation levels of total lymphocytes **(A)**, immunosuppressor levels **(B)**, immunostimulator levels **(C)** and MHC molecule levels **(D)** were calculated based on PSCA expression.
(DOCX)

**S8 Fig. Role of PSCA in the prognosis of LUAD. (A–B)** Univariate and multivariate Cox regression analyses of T stage, N stage, M stage, treatment outcomes, pathological stage and PSCA expression in LUAD; **(C)** Nomogram integrating PSCA expression and clinical variables such as age, sex, pathological stage, TNM stage and treatment outcomes; **(D)** KM curves demonstrating the effects of PSCA expression of prognosis in LUAD; **(E)** Calibration curve validating the prognostic significance of clinicopathological characteristics of tumours.
(DOCX)

**S9 Fig. ROC curves were plotted to validate the diagnostic efficacy of PSCA. (A)** Discrimination efficacy of PSCA for LUAD; **(B–F)** PSCA had better discrimination efficacy for TNM stage, residual tumour and primary treatment outcomes.
(DOCX)

**S10 Fig. Results of gene set enrichment analysis (GSEA) were summarised as enrichment maps. (A–F)** Pathway enrichment in the low-PSCA-expression group in the REACTOME database; **(G–L)** Pathway enrichment in the REACTOME, KEGG, PID, WP and BIO-CARTA databases was statistically significant in the 'c2.cp.KEGG.v7.1.symbols.gmt' gene set.
(DOCX)

**S1 Table. Validation of survival analysis in multiple databases.**
(DOCX)

**S2 Table. Univariate and multivariate regression analyses of the prognostic significance of PSCA in LUAD.**
(DOCX)

**S3 Table. Logistic regression model based on PSCA expression and clinicopathological features.**
(DOCX)

**S4 Table. Canonical pathways in MSigDB (c2.cp.v5.2.symbols.gmt) were used for GSEA.**
(DOCX)

## Author Contributions

**Conceptualization:** Chenxing Wang, Xingjia Zhu, Ming Zheng, Yixun Chen, Rui Jiang, Xinyi He, Zongheng Liu, Zhichao Lu, Ziheng Wang, Yang Yang.

**Data curation:** Chenxing Wang, Xingjia Zhu, Ming Zheng.

**Formal analysis:** Chenxing Wang, Xingjia Zhu, Zhichao Lu.

**Funding acquisition:** Yang Yang.

**Investigation:** Chenxing Wang, Xingjia Zhu, Ming Zheng, Yixun Chen.

**Methodology:** Chenxing Wang, Xingjia Zhu, Ming Zheng, Yixun Chen, Rui Jiang, Zhichao Lu.

**Project administration:** Ziheng Wang, Yang Yang.

**Resources:** Xingjia Zhu, Ziheng Wang.

**Software:** Chenxing Wang, Xingjia Zhu, Xinyi He, Zongheng Liu, Zhichao Lu.

**Supervision:** Ziheng Wang, Yang Yang.

**Validation:** Chenxing Wang, Ming Zheng, Xinyi He.

**Visualization:** Chenxing Wang, Xingjia Zhu, Ming Zheng, Rui Jiang, Zongheng Liu, Zhichao Lu.

**Writing – original draft:** Chenxing Wang, Ming Zheng, Yixun Chen, Ziheng Wang.

**Writing – review & editing:** Chenxing Wang, Xingjia Zhu, Ming Zheng, Yixun Chen, Rui Jiang, Xinyi He, Zongheng Liu, Zhichao Lu, Ziheng Wang, Yang Yang.

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
