## [Decision Letter · Decision Letter 0]

24 Jul 2023

PONE-D-23-16329Pan-cancer analysis of PSCA, which is associated with immune infiltration and affects patient prognosisPLOS ONE

Dear Dr. Wang,

Thank you for submitting your manuscript to PLOS ONE. After careful consideration, we feel that it has merit but does not fully meet PLOS ONE’s publication criteria as it currently stands. Therefore, we invite you to submit a revised version of the manuscript that addresses the points raised during the review process.

We look forward to receiving your revised manuscript.

Kind regards,

Zhijie Xu

Academic Editor

PLOS ONE

Journal Requirements:

"This study was supported by Science and Technology Project of Nantong City (MS22021044)."

"This study was supported by Science and Technology Project of Nantong City (MS22021044). The author YY was funded by the project. The funder's website is http://kjj.nantong.gov.cn/

4. Please upload a new copy of Figures 2 and 7 as the detail is not clear. Please follow the link for more information: " ext-link-type="uri" xlink:type="simple">https://blogs.plos.org/plos/2019/06/looking-good-tips-for-creating-your-plos-figures-graphics/"
" ext-link-type="uri" xlink:type="simple">https://blogs.plos.org/plos/2019/06/looking-good-tips-for-creating-your-plos-figures-graphics/"

Reviewers' comments:

Reviewer's Responses to Questions

**Comments to the Author**

1. Is the manuscript technically sound, and do the data support the conclusions?

Reviewer #1: Yes

Reviewer #2: Yes

2. Has the statistical analysis been performed appropriately and rigorously? 

Reviewer #1: Yes

Reviewer #2: Yes

3. Have the authors made all data underlying the findings in their manuscript fully available?

Reviewer #1: Yes

Reviewer #2: Yes

4. Is the manuscript presented in an intelligible fashion and written in standard English?

Reviewer #1: No

Reviewer #2: Yes

5. Review Comments to the Author

Reviewer #1: This manuscript titled “Pan-cancer analysis of PSCA, which is associated with immune infiltration and affects patient prognosis” presented a nice study with comprehensive design but the writing has some sever problems.

The followings comments and suggestions must be undertaken to improve the manuscript for readers.

1.In the abstract section “TME” abbreviation need complete description first.

2.In the Materials and Methods section, the Transwell assay needs to be described in detail, for example the abbreviation "SFM" needs to be described in full.

3.The statistical methods section should summarize the software used (Example: All procedures were accomplished with the aid of the Strawberry Perl (v. 5.30.0) and R (v. 4.3.0) programs.)

4.In the Materials and Methods section, the statistical analyses are not sufficiently described, and a complete summary of the statistical methods in the text is needed.

5.The figure notes are incorrectly labeled, for example, the result part (Fig. 1) should be (Fig. 1A). Supplementary Fig. 3F is missing and Fig. 3B, Fig. 3E is incorrectly labeled.

6.The language and logic of the manuscript still need further review.

Reviewer #2: 1.Provide more background information: Some sections, such as the diagnostic and prognostic potential of PSCA based on expression patterns, could be provided with more background information to help readers unfamiliar with the field to understand. Also, studies on the differences in PSCA expression in different cancers are recommended to provide more references or data to support them.

2.The article can be further organized and optimized in its structure to make the topics and subtopics clearer. For example, you may want to begin by stating the main objectives and methods of the study. Then, describe the various methods used in the research process in detail separately so that the reader can better understand the full picture of the study.

3.Refine your discussion: Although you have discussed a large amount of data and results, it is still possible to further refine your discussion as you explain the significance and potential clinical applications of these results.

4.The language expression of the article has some incoherent points. For example, the sentence "Through from the angle of cancer, the Prostate Stem Cell Antigen (PSCA) failure" has some grammatical and expression problems. The meaning of this sentence seems to be looking at the failure of PSCA from the angle of cancer, but this is not clear. The flow of language throughout the text needs to be further improved to allow the reader to better understand your point of view.

6. PLOS authors have the option to publish the peer review history of their article (what does this mean?). If published, this will include your full peer review and any attached files.

Reviewer #1: No

Reviewer #2: No

---

## [Author Response · Author response to Decision Letter 0]

29 Aug 2023

Dear editor 

Thank you very much for your interest in our manuscript entitled “Pan-cancer analysis of PSCA, which is associated with immune infiltration and affects patient prognosis”. To address the comments and concerns the Editor and Reviewers have raised, we have included a point by point response to each comment. Within the revised manuscript, changes to the text in response to the comments are marked as blue font. 

We appreciate the suggestions and comments by the Editor and the Reviewers. As a consequence of these valuable suggestions, we believe that our manuscript has been much strengthened.

Please feel free to contact me if you and editors have further suggestions. We would be happy to make a revision according to their comments. 

Kindly,

Chenxing Wang

Response letter titled “Pan-cancer analysis of PSCA, which is associated with immune infiltration and affects patient prognosis”

Q1: Please ensure that your manuscript meets PLOS ONE's style requirements, including those for file naming. 

Response: Dear editor, thank you for your suggestion. We have revised our manuscript to comply with PLOS ONE's style requirements and invite you to review it.

 

Q2: Thank you for stating the following in the Acknowledgments Section of your manuscript: "This study was supported by Science and Technology Project of Nantong City (MS22021044)." We note that you have provided funding information that is not currently declared in your Funding Statement. However, funding information should not appear in the Acknowledgments section or other areas of your manuscript. We will only publish funding information present in the Funding Statement section of the online submission form. Please remove any funding-related text from the manuscript and let us know how you would like to update your Funding Statement. Currently, your Funding Statement reads as follows: "This study was supported by Science and Technology Project of Nantong City (MS22021044). The author YY was funded by the project. The funder's website is http://kjj.nantong.gov.cn/. The funders had no role in study design, data collection and analysis, decision to publish, or preparation of the manuscript." Please include your amended statements within your cover letter; we will change the online submission form on your behalf.

Response: Dear Editor, thank you for your suggestion. We apologise that we incorrectly presented funding information in the manuscript acknowledgements. We have removed this information from the manuscript. Please review it.

 

Q3: PLOS ONE now requires that authors provide the original uncropped and unadjusted images underlying all blot or gel results reported in a submission’s figures or Supporting Information files. This policy and the journal’s other requirements for blot/gel reporting and figure preparation are described in detail at https://journals.plos.org/plosone/s/figures#loc-blot-and-gel-reporting-requirements and https://journals.plos.org/plosone/s/figures#loc-preparing-figures-from-image-files. When you submit your revised manuscript, please ensure that your figures adhere fully to these guidelines and provide the original underlying images for all blot or gel data reported in your submission. See the following link for instructions on providing the original image data: https://journals.plos.org/plosone/s/figures#loc-original-images-for-blots-and-gels. 

Response: Dear Editor, Thank you for your suggestion. We apologise that we forgot to upload the raw data for the western blot. We have now uploaded it as a new file named "TEST RAW DATA". Please check it. 

 

Q4: Please upload a new copy of Figures 2 and 7 as the detail is not clear. Please follow the link for more information: https://blogs.plos.org/plos/2019/06/looking-good-tips-for-creating-your-plos-figures-graphics/"https://blogs.plos.org/plos/2019/06/looking-good-tips-for-creating-your-plos-figures-graphics/" 

Response: Dear Editor, Thank you for your suggestion. We have re-uploaded Figures 2 and 7. Please check it.

 

Dear reviewer1 

Thank you very much for your interest in our manuscript entitled “Pan-cancer analysis of PSCA, which is associated with immune infiltration and affects patient prognosis”. To address the comments and concerns the Editor and Reviewers have raised, we have included a point by point response to each comment. Within the revised manuscript, changes to the text in response to the comments are marked as blue font. 

We appreciate the suggestions and comments by the Editor and the Reviewers. As a consequence of these valuable suggestions, we believe that our manuscript has been much strengthened.

Please feel free to contact me if you and editors have further suggestions. We would be happy to make a revision according to their comments. 

Kindly,

Chenxing Wang

 

Response letter titled “Pan-cancer analysis of PSCA, which is associated with immune infiltration and affects patient prognosis”

Q1: In the abstract section “TME” abbreviation need complete description first.

Response: Dear Reviewer, your comments are very informative. In order for the reader to better understand the content of the article, we have provided a detailed description of "TME" based on your suggestions.

 

Q2: In the Materials and Methods section, the Transwell assay needs to be described in detail, for example the abbreviation "SFM" needs to be described in full.

Response: Dear Reviewer, your suggestion is pertinent, and we have refined and modified the experimental methodology on Transwell Assay for better understanding of the readers.

 

Q3: The statistical methods section should summarize the software used (Example: All procedures were accomplished with the aid of the Strawberry Perl (v. 5.30.0) and R (v. 4.3.0) programs.)

Response: Dear Reviewer, Thank you for your suggestion. The relevant software versions have been indicated in the article.

 

Q4: In the Materials and Methods section, the statistical analyses are not sufficiently described, and a complete summary of the statistical methods in the text is needed.

Response: Dear Reviewer. Thank you for your suggestion. We have summarized the statistical methods used in the article and supplemented them in the Methods section of the paper.

 

Q5: The figure notes are incorrectly labeled, for example, the result part (Fig. 1) should be (Fig. 1A). Supplementary Fig. 3F is missing and Fig. 3B, Fig. 3E is incorrectly labeled.

Response: Dear Reviewer. Thank you for your suggestion. We have revised the relevant figure captions and checked the captions in other parts of the article.

 

Q6: The language and logic of the manuscript still need further review.

Response: Dear Reviewer, In accordance with your request, we sent the manuscript of the article to a professional polishing company and had it polished by native English speakers, the polish report of which we have attached.

 

Dear reviewer2 

Thank you very much for your interest in our manuscript entitled “Pan-cancer analysis of PSCA, which is associated with immune infiltration and affects patient prognosis”. To address the comments and concerns the Editor and Reviewers have raised, we have included a point by point response to each comment. Within the revised manuscript, changes to the text in response to the comments are marked as blue font. 

We appreciate the suggestions and comments by the Editor and the Reviewers. As a consequence of these valuable suggestions, we believe that our manuscript has been much strengthened.

Please feel free to contact me if you and editors have further suggestions. We would be happy to make a revision according to their comments. 

Kindly,

Chenxing Wang

 

Response letter titled “Pan-cancer analysis of PSCA, which is associated with immune infiltration and affects patient prognosis”

Q1: Provide more background information: Some sections, such as the diagnostic and prognostic potential of PSCA based on expression patterns, could be provided with more background information to help readers unfamiliar with the field to understand. Also, studies on the differences in PSCA expression in different cancers are recommended to provide more references or data to support them.

Response: Dear reviewer, Regarding the background of the study on the differences in PSCA expression in different cancers, our review of the literature revealed that PSCA expression is elevated in pancreatic and gastric cancers, and there are fewer references exploring the relationship between PSCA and other cancers as well as the prognostic potential, and thus we this study can further complement the link between PSCA and cancer.

1. Teng KY, Mansour AG, Zhu Z, Li Z, Tian L, Ma S, Xu B, Lu T, Chen H, Hou D, Zhang J, Priceman SJ, Caligiuri MA, Yu J. Off-the-Shelf Prostate Stem Cell Antigen-Directed Chimeric Antigen Receptor Natural Killer Cell Therapy to Treat Pancreatic Cancer. Gastroenterology. 2022 Apr;162(4):1319-1333. pii: S0016-5085(22)00001-4. doi: 10.1053/j.gastro.2021.12.281. PubMed PMID: 34999097.

2. Hess T, Maj C, Gehlen J, Borisov O, Haas SL, Gockel I, Vieth M, Piessen G, Alakus H, Vashist Y, Pereira C, Knapp M, Schüller V, Quaas A, Grabsch HI, Trautmann J, Malecka-Wojciesko E, Mokrowiecka A, Speller J, Mayr A, Schröder J, Hillmer AM, Heider D, Lordick F, Pérez-Aísa Á, Campo R, Espinel J, Geijo F, Thomson C, Bujanda L, Sopeña F, Lanas Á, Pellisé M, Pauligk C, Goetze TO, Zelck C, Reingruber J, Hassanin E, Elbe P, Alsabeah S, Lindblad M, Nilsson M, Kreuser N, Thieme R, Tavano F, Pastorino R, Arzani D, Persiani R, Jung JO, Nienhüser H, Ott K, Schumann RR, Kumpf O, Burock S, Arndt V, Jakubowska A, Ławniczak M, Moreno V, Martín V, Kogevinas M, Pollán M, Dąbrowska J, Salas A, Cussenot O, Boland-Auge A, Daian D, Deleuze JF, Salvi E, Teder-Laving M, Tomasello G, Ratti M, Senti C, De Re V, Steffan A, Hölscher AH, Messerle K, Bruns CJ, Sīviņš A, Bogdanova I, Skieceviciene J, Arstikyte J, Moehler M, Lang H, Grimminger PP, Kruschewski M, Vassos N, Schildberg C, Lingohr P, Ridwelski K, Lippert H, Fricker N, Krawitz P, Hoffmann P, Nöthen MM, Veits L, Izbicki JR, Mostowska A, Martinón-Torres F, Cusi D, Adolfsson R, Cancel-Tassin G, Höblinger A, Rodermann E, Ludwig M, Keller G, Metspalu A, Brenner H, Heller J, Neef M, Schepke M, Dumoulin FL, Hamann L, Cannizzaro R, Ghidini M, Plaßmann D, Geppert M, Malfertheiner P, Gehlen O, Skoczylas T, Majewski M, Lubiński J, Palmieri O, Boccia S, Latiano A, Aragones N, Schmidt T, Dinis-Ribeiro M, Medeiros R, Al-Batran SE, Leja M, Kupcinskas J, García-González MA, Venerito M, Schumacher J. Dissecting the genetic heterogeneity of gastric cancer. EBioMedicine. 2023 Jun;92:104616. doi: 10.1016/j.ebiom.2023.104616. Epub 2023 May 18. Erratum in: EBioMedicine. 2023 Aug;94:104709. PMID: 37209533; PMCID: PMC10212786.

 

Q2: The article can be further organized and optimized in its structure to make the topics and subtopics clearer. For example, you may want to begin by stating the main objectives and methods of the study. Then, describe the various methods used in the research process in detail separately so that the reader can better understand the full picture of the study.

Response: Dear reviewer, thank you for your sincere suggestion. We have optimised the article structure according to your request. And, the revised manuscript we have handed over to native speakers for appropriate touch-ups. We kindly ask you to review the revised manuscript.

 

Q3: Refine your discussion: Although you have discussed a large amount of data and results, it is still possible to further refine your discussion as you explain the significance and potential clinical applications of these results.

Response: Dear reviewer, thank you for your informative suggestions, we have revised and improved the discussion to focus on the importance of our findings and potential clinical applications.

 

Q4: The language expression of the article has some incoherent points. For example, the sentence "Through from the angle of cancer, the Prostate Stem Cell Antigen (PSCA) failure" has some grammatical and expression problems. The meaning of this sentence seems to be looking at the failure of PSCA from the angle of cancer, but this is not clear. The flow of language throughout the text needs to be further improved to allow the reader to better understand your point of view.

Response: Dear reviewer, we did have problems with the English of our manuscript, which may have led to misunderstandings among readers. Based on your suggestion, we submitted the article to a professional polishing agency and had it revised by native speakers and issued a revision report, which we have attached below.

---

## [Decision Letter · Decision Letter 1]

25 Jan 2024

Pan-cancer analysis of PSCA that is associated with immune infiltration and affects patient prognosis

PONE-D-23-16329R1

Dear Dr. Wang,

We’re pleased to inform you that your manuscript has been judged scientifically suitable for publication and will be formally accepted for publication once it meets all outstanding technical requirements.

Kind regards,

Zhijie Xu

Academic Editor

PLOS ONE

Additional Editor Comments (optional):

It is a good revision, and story.

Reviewers' comments:

Reviewer's Responses to Questions

**Comments to the Author**

1. If the authors have adequately addressed your comments raised in a previous round of review and you feel that this manuscript is now acceptable for publication, you may indicate that here to bypass the “Comments to the Author” section, enter your conflict of interest statement in the “Confidential to Editor” section, and submit your "Accept" recommendation.

Reviewer #1: All comments have been addressed

Reviewer #2: All comments have been addressed

2. Is the manuscript technically sound, and do the data support the conclusions?

Reviewer #1: Yes

Reviewer #2: Yes

3. Has the statistical analysis been performed appropriately and rigorously? 

Reviewer #1: Yes

Reviewer #2: Yes

4. Have the authors made all data underlying the findings in their manuscript fully available?

Reviewer #1: Yes

Reviewer #2: Yes

5. Is the manuscript presented in an intelligible fashion and written in standard English?

Reviewer #1: Yes

Reviewer #2: Yes

6. Review Comments to the Author

Reviewer #1: The authors did a great job of answering my questions. I don't have any additional comments to make, congratulations to the authors.

Reviewer #2: The manuscript, as evaluated, excels in presenting comprehensive and detailed data. The provided information substantially bolsters the drawn conclusions, reflecting a commendable alignment between data and outcomes. The statistical methods employed exhibit appropriateness and precision, contributing to the robustness of the findings. The overall quality of the data analysis enhances the credibility of the research, reinforcing its scientific merit. Based on the thoroughness of data presentation, the robust support for conclusions, and the appropriateness of statistical methods, I recommend accepting this manuscript for publication.

7. PLOS authors have the option to publish the peer review history of their article (what does this mean?). If published, this will include your full peer review and any attached files.

Reviewer #1: No

Reviewer #2: No

---

## [Editor Report · Acceptance letter]

26 Apr 2024

PONE-D-23-16329R1 

PLOS ONE

Dear Dr. Wang, 

I'm pleased to inform you that your manuscript has been deemed suitable for publication in PLOS ONE. Congratulations! Your manuscript is now being handed over to our production team.

Kind regards, 

on behalf of

Prof. Zhijie Xu 

Academic Editor

PLOS ONE